# CAIX Regulates Invadopodia Formation through Both a pH-Dependent Mechanism and Interplay with Actin Regulatory Proteins

**DOI:** 10.3390/ijms20112745

**Published:** 2019-06-04

**Authors:** Michaela Debreova, Lucia Csaderova, Monika Burikova, Lubomira Lukacikova, Ivana Kajanova, Olga Sedlakova, Martin Kery, Juraj Kopacek, Miriam Zatovicova, Jozef Bizik, Silvia Pastorekova, Eliska Svastova

**Affiliations:** 1Biomedical Research Center, Institute of Virology, Department of Cancer Biology, Slovak Academy of Sciences, Dubravska cesta 9., 845 05 Bratislava, Slovakia; mdebreova@gmail.com (M.D.); lucia.csaderova@savba.sk (L.C.); lubomira.lukacikova@savba.sk (L.L.); iva.kajanova@gmail.com (I.K.); viruolse@savba.sk (O.S.); mato.kery@gmail.com (M.K.); virukopa@savba.sk (J.K.); viruzato@savba.sk (M.Z.); virusipa@savba.sk (S.P.); 2Biomedical Research Center, Cancer Research Institute, Department of Molecular Oncology, Slovak Academy of Sciences, Dubravska cesta 9., 845 05 Bratislava, Slovakia; burikovamonika@gmail.com (M.B.); Jozef.Bizik@savba.sk (J.B.)

**Keywords:** carbonic anhydrase IX, invadopodia, matrix degradation, CAIX targeting antibodies, chorioallantoic membranes, lungs colonization

## Abstract

Tumor metastasis is tightly linked with invasive membrane protrusions, invadopodia, formed by actively invading tumor cells. Hypoxia and pH modulation play a role in the invadopodia formation and in their matrix degradation ability. Tumor-associated carbonic anhydrase IX (CAIX), induced by hypoxia, is essential for pH regulation and migration, predisposing it as an active component of invadopodia. To investigate this assumption, we employed silencing and inhibition of CA9, invadopodia isolation and matrix degradation assay. Quail chorioallantoic membranes with implanted tumor cells, and lung colonization assay in murine model were used to assess efficiency of *in vivo* invasion and the impact of CAIX targeting antibodies. We showed that CAIX co-distributes to invadopodia with cortactin, MMP14, NBCe1, and phospho-PKA. Suppression or enzymatic inhibition of CAIX leads to impaired invadopodia formation and matrix degradation. Loss of CAIX attenuated phosphorylation of Y421-cortactin and influenced molecular machinery coordinating actin polymerization essential for invadopodia growth. Treatment of tumor cells by CAIX-specific antibodies against carbonic or proteoglycan domains results in reduced invasion and extravasation *in vivo*. For the first time, we demonstrated *in vivo* localization of CAIX within invadopodia. Our findings confirm the key role of CAIX in the metastatic process and gives rationale for its targeting during anti-metastatic therapy.

## 1. Introduction

The expansion of tumor cells from their primary residence to distant organs in the body is associated with the extracellular matrix (ECM) degradation and dynamic interactions between the ECM and the cell cytoskeleton [1]. Actively migrating and invading tumor cells rely on the occurrence of plasma membrane protrusions on their ventral side, termed invadopodia [2]. These actin-rich cell structures intensively digest the ECM barrier allowing the infiltration of healthy tissues. Extracellular acidification is essential for invadopodial function through the activation of matrix metalloproteinases [3]. On the other hand, alkalinization of intracellular pH (pHi) leads to the production of the free-barbed actin ends supporting actin polymerization and invadopodia elongation [4]. Thus, pH regulators generating pHe and pHi gradients are critical for invadopodia function and tumor invasion.

There are several regulatory mechanisms controlling the activity of invadopodia. The data indicating the role of hypoxia and pH modulation in the invadopodia formation and functioning have been increasing [3,5,6,7,8]. Carbonic anhydrase IX (CAIX) is a hypoxia-induced, cancer-associated protein, the expression of which correlates with aggressive behavior and metastases [9,10,11]. Overexpression of CAIX in tumors is a part of the survival strategy during adaptation to hypoxia. CAIX is a transmembrane enzyme catalyzing the reversible hydration of carbon dioxide to bicarbonate ions and protons [12,13]. Through its enzymatic activity, CAIX contributes to the acidification of extracellular space and generation of the pro-invasive tumor microenvironment [12,13,14]. In the lamellipodia of migrating cells, CAIX forms a transport metabolon with bicarbonate transporters (AE2, NBCe1), enhancing the ion flux through the plasma membrane, and promoting cell migration [14]. The mechanism behind the hypoxic activation of CAIX involves elevated cAMP levels mediating the activation of the protein kinase A (PKA). This phosphorylates CAIX at Thr443, located in its intracellular part, which stimulates enzymatic activity of this protein [15]. Recently, it was demonstrated that CAIX associates with matrix metalloprotease MMP14 via Thr443, and protons generated by CAIX are required for MMP14 activity [16]. Importantly, measurements of extracellular pH confirmed that CAIX acidifies the tumor microenvironment *in vivo* and maintains pHe acidity at values favoring cancer cell invasion and metastasis [17].

Generation of focalized pH nanodomains and invadopodia function depend on Na^+^/H^+^ exchanger 1 (NHE1) [3,4,18,19]. During invadopodia maturation, NHE1 is recruited and drives extracellular acidification, promoting ECM proteolysis and local intracellular alkalization. Increased pHi disrupts cortactin-cofilin binding, thus releasing cofilin for actin-severing activity essential for invadopodia growth [4,20]. It was shown that cofilin acts as a pH sensor mediating pH-dependent actin filament dynamics [21]. Cortactin phosphorylation is a master regulator of invadopodia maturation. Tyrosine kinases of the Src- and Abl-families localize to invadopodia precursors, and through the cortactin phosphorylation facilitate the assembly of Nck1-WASP-Arp2/3 signaling complex [20,22,23]. Cortactin phosphorylation of tyrosines Y421 and Y466 controls cofilin and Arp2/3 complex-dependent actin polymerization [20]. Besides the release of cofilin, pY421 and pY466 of cortactin are essential for binding of Nck1, which recruits the N-WASP-Arp2/3 complex. Abrogation of either phoshotyrosine 421 or 466 causes almost complete inhibition of actin polymerization in invadopodia [24]. Importantly, cortactin tyrosine phosphorylation mediates NHE1 recruitment, which subsequently affects cortactin-cofilin interaction in a pH-dependent manner [4].

Furthermore, voltage gated-sodium channel NaV1.5, which also associates with NHE1 in invadopodia, promotes ECM degradation and remodeling in high-grade breast cancers [25]. Besides the regulation of NHE1 exchanger, NaV1.5 also enhances Src kinase activity and cortactin phosphorylation on Y421. This specific phosphorylation disturbs cortactin-cofilin interaction essential for F-actin polymerization in invadopodia [8]. Several invasive tumor subtypes have been shown to utilize invadopodia during invasion, including breast, head and neck, colon, pancreas, and prostate carcinomas [26,27]. It was confirmed that circulating tumor cells attached on capillaries form protrusions that cross the endothelial layer into the extravascular stroma [28]. These protrusions are classified as invadopodia since they are positive for invadopodial markers cortactin, MMP14, Tks4 and Tks5. Silencing of cortactin and Tks proteins dramatically inhibits cancer cell extravasation [29]. Thus, the utilization of invadopodia by circulating tumor cells to penetrate the secondary organs and establish metastasis is a general feature of cancer.

In this paper, we investigated mechanisms, by which CAIX regulates invadopodia formation, maturation, and subsequent matrix degradation and cell invasion. Our data show that CAIX influences invadopodia-related events by its expression level as well as by the correlated catalytic function. In addition, we demonstrated the role of CAIX in tumor cell invasion and extravasation *in vivo* through quail embryo model and murine lungs colonization model. Our analyses have also shown that CAIX targeting by specific monoclonal antibodies causes a significant inhibition of *in vivo* tumor cell invasion. These results confirm a key role of the CAIX protein in the metastatic process and suggest a basis for its targeting during anti-metastatic therapy.

## 2. Results

### 2.1. The CAIX Protein Distributes to Proteolytically Active Invadopodia

Since the CAIX protein is known to be involved in pH regulation, migration, and focal adhesion, we investigated the subcellular localization of CAIX during 3D invasion. We examined colocalization of CAIX with invadopodia markers cortactin and F-actin. As soon as 5 hrs after the seeding of the hypoxia-preincubated cells, we detected codistribution of CAIX with cortactin in invadopodia precursors characterized by accumulation of cortactin at the ventral surface of cells (Figure 1A). Then, 24 hrs after the seeding on collagen, CAIX colocalized with F-actin in protruding invadopodia where actin-polymerization occurs (Figure 1B upper part – xy sections, 1B lower part – xz sections).

Progression of invadopodia requires the focalized degradation of the ECM. We analyzed the colocalization of CAIX protein with the marker of proteolytic activity DQ-Red BSA in C33 cells. The cells were cultured on Matrigel substrates mixed with quenched DQ-Red BSA, which emits the fluorescence signal only after proteolytic cleavage. Using a confocal microscope with Z-stack scanning, direct colocalization of CAIX with focal proteolysis of the ECM was detected in proteolytically active invadopodia (Figure 1C,D).

To ascertain CAIX in mature invadopodia penetrating the matrix, HT1080 cells were osmotically lysed and incubated on gelatin layer for 48 h in hypoxia. Then, we either separated proteins into cell bodies fraction and invadopodial fraction to evaluate the protein composition by Western blot, or directly immunofluorescently stained the invadopodia that remained stuck in the gelatin. The z-stack analysis of CAIX and cortactin distribution showed their colocalization within deep gelatin layers corresponding to invadopodia entrapped in the ECM (Figure 1E). To verify that the observed fluorescent signal came specifically from the separated invadopodia, and not from the cell bodies, nuclei were stained with DAPI and images were taken in transmitted and fluorescent light microscopy. Pictures were negative for the blue signal and no intact cells were detected by transmitted light microscopy (data not shown). Western blot analysis confirmed CAIX as a component of the invadopodial fraction (Figure 1F) together with known invadopodial markers cortactin and MMP14. As we have shown previously, enzymatic activity and acidification capability of CAIX are regulated by PKA, which is also stimulated by hypoxia [15]. Moreover, the PKA-gated Rho-A signal module regulating NHE1-dependent invasion is located in the leading edge of pseudopodia, and is enhanced by hypoxia [18]. Importantly, we detected an active form of PKA in invadopodia (p-Thr197 PKA), supporting the view that CAIX restricted to invadopodia is catalytically active.

### 2.2. Bicarbonate Transporter NBCe1 Colocalizes with Cortactin and CAIX in Invadopodia

Invadopodia formation is potentiated by local intracellular alkalization and extracellular acidification. We have previously shown that CAIX cooperates with bicarbonate transporters in lamellipodia to form transport the metabolon, which enhances the efficiency of ion transport through the plasma membrane. Therefore, we investigated whether CAIX is involved in pH regulation within the invadopodial compartment and whether its association with bicarbonate transporter NBCe1 is also maintained in invadopodia as their cooperation may result in the generation of local pH nanodomains affecting invadopodia formation and maturation. Hypoxia pre-incubated HT1080 cells were seeded on collagen and, subsequently after 5 h of incubation, were fixed, and double-stained for CAIX and NBCe1, or cortactin and NBCe1. We showed that sodium bicarbonate cotransporter NBCe1 is localized to the cortactin-rich perinuclear zone, characteristic of invadopodia precursors (Figure 2). Pearson’s correlation coefficient *r* = 0.8 (in invadopodial perinuclear zone) confirmed a high positive correlation between cortactin and NBCe1. CAIX and NBCe1 displayed the same codistribution in the invadopodial area. In accordance with the known localization of cortactin, NBCe1, and CAIX in lamellipodia, we also detected their distribution in these cell structures.

### 2.3. Suppression of CAIX Reduces Invadopodia Formation

As CAIX acidifies the extracellular pH and cooperates with bicarbonate transporters contributing to intracellular alkalization, we were interested in consequences of CAIX impairment with respect to invadopodia formation. Hypoxic, CAIX-expressing control cells (siCtrl) and CA9-silenced HT1080 cells (siCA9) were grown for 5h on FITC-collagen. Transient silencing efficiency was approximately 90% of the CAIX protein level (Figure 3A). CAIX positive cells (siCtrl) showed a higher number of cortactin-composed invadopodia characterized by a stronger fluorescent intensity in comparison to CA9-suppressed cells (siCA9) (Figure 3B). The area covered by cortactin-stained invadopodia was reduced from around 12% in CAIX-expressing cells to only 6% in CA9-silenced cells (Figure 3C). Efficiency of the matrix cleavage as a consequence of focalized extracellular acidification was analyzed using DQ-Red BSA mixed with Matrigel (Figure 3D–I). We compared the ECM digestion in hypoxia pre-incubated CAIX-expressing versus CA9-silenced HeLa cells. After 24 h of ECM degradation in 2% hypoxia, the number and intensity of Matrigel-digested spots was evaluated. Confocal fluorescence microscope images showing proteolytic activity of MMPs in Matrigel as the red signal superimposed with the phase-contrast image clearly demonstrate that silencing of CAIX strongly attenuates the proteolytic cleavage of matrix (Figure 3D). Analysis of data in the histogram revealed that CAIX-positive cells display a higher number of spots of very efficient digestion, indicating more potent matrix degradation than in CA9-silenced cells (Figure 3E). Using the z-stack analysis by a confocal microscope, we quantified local proteolytic activity separately in different layers of protruding invadopodia (Figure 3F). We started at the level of cell attachment to the substrate and continued downward into the matrix. Silencing of CA9 reduced proteolytic activity at all planes of hypoxic invadopodia. The difference in the ECM digestion efficiency of both types of cells is illustrated by a 3D projection image of the degraded matrix signal (Figure 3G). Overall, CAIX-positive cells degraded the matrix 67% more efficiently than CA9-silenced cells (Figure 3H).

Furthermore, we proved that the enzymatic activity of CAIX is essential for efficient Matrigel cleavage, since its inhibition by specific inhibitor HSFA (4-Homosulfanilamide) severely decreases the number of regions with intense ECM proteolysis compared to cells with catalytically active CAIX (Figure 3I).

### 2.4. Loss of CAIX Decreases Levels of Invadopodia Components and Signaling

Anticipated role of CAIX in invadopodia formation was originally based on its influence on pHe and pHi and its lamellipodial localization. Guided by our previously published microarray data [30], we decided to investigate a possible influence of CAIX expression on actin organizing proteins associated with invadopodia. We detected a decreased level of Arp2 protein essential for actin nucleation and branching during invadopodia elongation in HT1080 and HeLa cells with silenced CA9. Western blot analysis also confirmed a reduction of integrin β3 in HT1080 cell line after CAIX suppression. We did not detect integrin β3 protein in HeLa cells, consistent with Genevestigator analysis revealing low expression of the ITGB3 gene in the HeLa cell line (data not shown).

The total cortactin level was not changed in CA9-silenced HT1080 or HeLa cells but phosphorylation of cortactin on tyrosine Y421 was decreased, relative to their control counterparts (Figure 4A).

Generation of invadopodia and subsequent matrix cleavage is thought to promote cancer cell invasion through the ECM. Functional consequences of CA9-silencing in this process were examined in the invasion assay using xCELLigence RTCA DP Instrument (Figure 4B). Hypoxia pre-incubated siCtrl and siCA9 HT1080 cells were applied on Boyden chamber in which the membranes were pre-coated with collagen. Silencing of CA9 reduced the invasion of HT1080 cells by about 25%.

Together, these results indicate an interplay between CAIX and actin-associated proteins during invadopodia formation demonstrated in the effect of CAIX on the expression and signaling of actin-regulating proteins.

### 2.5. Treatment of Tumor Cells with CAIX Targeting Antibodies Suppresses Their Metastatic Properties

To monitor the metastatic properties of cancer cells and to study the effect of potential therapeutic anti-CAIX antibodies *in vivo*, we used the chorioallantoic membrane (CAM) invasion assay. We optimized this model for the epithelial cell line derived from well differentiated human squamous cell carcinoma of the esophagus TE-1. In TE-1 cells, CAIX is induced by hypoxia with basolateral localization on the cell surface.

Hypoxia pre-incubated TE-1 cells (48 h, 1% O_2_) were trypsinized and, in indicated samples, they were mixed with specific anti-CAIX antibodies Ab10 or M75, which recognize the catalytic (CA) or PG-domain of CAIX protein, respectively. Cells were grafted onto 7-day-old quail embryo and cultured *ex ovo* for 3 days. Interestingly, CAIX-positive cells adhered directly on the ectoderm while the CAIX-negative subpopulation of TE-1 arranged on top of the CAIX-positive layer (Figure 5A). Control CAIX expressing cells without anti-CAIX treatment efficiently invaded into the mesoderm and formed metastases (Figure 5A,B). Cells expressing CAIX are positioned at the invasive front of migrating clusters (Figure 6B, left). In Figure 5B (right) migratory morphology of CAIX-positive tumor cells prepared to invade the ectoderm is visible. Invadopodium with accumulated CAIX at its tip can be seen protruding deep into the mesoderm. To our knowledge, this is the first demonstration of CAIX distribution into *in vivo* invadopodia. In addition, anti-CA monoclonal antibody Ab10 completely inhibited metastatic properties of cancer cells, when compared to the control TE-1 samples (Figure 5C, left). Anti-PG antibody, M75, strongly suppressed the invasive capability of TE-1 cells which were able to pass through the ectoderm only in 30% of studied cases. No metastases were detected in the mesoderm layer. Evaluation of the degree of the invasion of TE-1 cells, control and antibodies-treated, into CAM layers is summarized in Figure 5D.

### 2.6. Effect of anti-CAIX Antibody Treatment on Experimental Lung Metastasis of HT1080 Cells

During extravasation, cancer cells form membrane protrusions enriched by MMP14 and cortactin characteristic for invadopodia. Given our observations in the quail embryo CAM model, we evaluated whether anti-CAIX antibodies can inhibit the formation of experimental metastases. Hypoxic HT1080-RFP cells with or without Ab10 or M75 treatment were injected into the tail vein of NMRI mice. Metastatic colonies in murine lungs were imaged ex vivo after 10 days using a Caliper imaging system. Total radiant efficiency reflects the amount of cancer cells in murine lungs. Pre-incubation of HT1080-RFP cells with Ab10 or M75, and subsequent administration of 2 doses of antibodies during 10 days (after tail vein injection) caused a marked decrease in lung colonization by these cells, as determined by fluorescence imaging of the RFP signal by IVIS (Figure 6). The *in vivo* data thus support the role of CAIX in circulating tumor cells and indicate a possible benefit of anti-CAIX therapy in attenuation of their extravasation and metastasis formation.

## 3. Discussion

The tumor microenvironment fundamentally affects the invasive capability of cells and, subsequently, tumor progression. Cellular adaptive program triggered by hypoxia via HIF1a includes enhanced expression of pH regulators like NHE1, CAIX, and MCT4 to maintain pHi homeostasis. Simultaneously, activity of these proteins generates extracellular acidosis contributing to effective matrix cleavage essential for tumor cell movement toward blood vessels. Actively migrating and invading metastatic cells rely on plasma membrane protrusions, invadopodia. NHE1-dependent acidification of the extracellular space surrounding invadopodia promotes the activity of MMPs and cathepsins to cleave the ECM and push the invasion [3,4,31]. Local intracellular alkalization in invadopodia tips is crucial for actin remodeling through the pH sensitive proteins, such as talin and cofilin, and invadopodia growth [4,21]. The hypoxic up-regulation of CAIX, and its well documented role in pH regulation, adhesion, migration, and invasion, predispose this tumor-associated protein to be an active component of invadopodia. In this article, we show that CAIX colocalizes with invadopodia markers cortactin and F-actin in intact cells, as well as in isolated gelatin-entrapped invadopodia. Silencing of CA9 reduces the invadopodia formation and matrix degradation. The total level of cortactin did not change in siCA9 HeLa or HT1080 cells, but the phosphorylation of cortactin at tyrosine 421 (Y421) was attenuated. Phosphorylation at Y421 is essential for Nck1 binding to cortactin and the subsequent Nck1–N-WASp–Arp2/3 complex assembly which is required for efficient actin polymerization in the invadopodia [20,24]. Moreover, cortactin-cofilin interaction in invadopodia is pH dependent, and the release of cofilin from cortactin binding (effective in higher pH values characteristic for invadopodia) is crucial for cofilin-mediated free-barbed end (FBE) formation [4,21]. Y421-cortactin phosphorylation is also important for NHE1 recruitment to invadopodia. Cells expressing the non-phosphorylatable cortactin mutant exhibited decreased pH inside invadopodia. Magalhaes and colleagues also showed that invadopodial pH values oscillate more than pH in the rest of cytoplasm, indicating higher pH-sensitivity of processes ongoing in invadopodia. Inhibition of CAIX catalytic activity by HSFA decreased the proteolytic degradation of the ECM at invadopodia, confirming the importance of pH regulating role of CAIX during invadopodia growth and matrix penetration. Thus, functional consequences of CAIX downregulation range from reduced invadopodia formation to impaired invasive ability of siCA9 cells. CAIX-related changes are summarized in the schematic model (Figure 7).

Previously, we revealed that CAIX forms a transport metabolon with bicarbonate transporters in lamellipodia of migrating cells. This localization as well as the presence of NHE1 in lamellipodia are essential for the generation of pHi and pHe gradients along the axis of translocation. Here, we show colocalization of NBCe1 with cortactin at perinuclearly accumulated invadopodia where CAIX colocalizes with NBCe1, and also cortactin. Recently, Boedtkjer et al. [32] showed that NBCn1 mediated Na^+^-HCO_3_^−^ cotransport creates pHi gradient along migrating cells, which promotes filopodia formation and migration. In cells from NBCn1 KO mice, the pHi gradient between the tip of filopodia and cytosol disappeared. We propose that CO_2_/HCO_3_^−^ buffering mediated by CAIX and Na^+^-HCO_3_^−^ cotransporters locally modifies pHi and pHe gradients at invadopodia, facilitating actin remodeling and invadopodial proteolytic activity. The relevance of CAIX enzymatic activity in invadopodia is also underlined by our observation that the active form of PKA (pThr197-PKA), which activates CAIX by its phosphorylation on Thr443 [15], is codistributed along with CAIX in isolated invadopodia. Interestingly, Swayampakula and colleagues [16] showed that phosphorylation of Thr443 on CAIX is essential for its interaction with MMP14 and protons produced by CAIX catalytic activity are critical for MMP14-mediated collagen degradation. On the basis of these findings we suggest that enzymatically active CAIX regulated by PKA plays an essential role in proteolyticaly active invadopodia.

To determine whether treatment by antibodies against CAIX modifies the propensity of cancer cells to invade and disseminate *in vivo*, the chorioallantoic membranes of quail embryos were used for invasion assay. Interestingly, the CAIX positive subpopulation of hypoxic TE-1 cells layered directly on the ectoderm and constituted the invasive front. Binding of the M75 antibody to the CAIX PG-domain inhibited infiltration of cancer cells. We assume that interactions mediated by the CAIX PG-domain influence the effectiveness of invasion. It has been reported that CAIX associates with β1 and α2 integrin subunits [16]. Integrin interactions with different proteins e.g., uPAR or emmprin, play the role in the invasive ability of tumor cells [31,32,33]. Thus, it is conceivable that disruption of the interaction between CAIX and integrins can inhibit integrin function and integrin-mediated invasion.

Importantly, invadopodia developed by cancer cells projected into CAM were strongly positive for CAIX. This represents the first *in vivo* evidence of CAIX localization in invadopodia. Treatment of TE-1 cells with antibody against CAIX catalytic domain completely abrogated the metastatic ability of cancer cells. We assume that binding of Ab10 to the catalytic domain either inhibits the enzymatic activity of CAIX or disrupts CAIX-bicarbonate transporters metabolon, thus reducing the generation of acidic pH-nanodomains, which are crucial for MMP activity and cells invasion.

Extravasation step of cancer cells during a metastatic cascade also depends on the invadopodia formation. Circulating tumor cells adhering on the endothelial layer extend protrusions, which are rich in invadopodia markers MMP14, cortactin, Tks4, and Tks5 [29]. Cortactin depletion as well as silencing of Tks proteins abrogate invadopodia formation, and consequently decreases extravasation in the CAM model. In our experiment, pre-incubation of hypoxic HT1080-RFP cells with anti-CAIX antibody targeting PG- or CA-domain reduced the number of mouse lung metastases. In this short-term experiment, metastatic colonies were evaluated 10 days after tail vein inoculation, which means that reduced extravasation was a main factor behind decreased metastasis formation. Our results correlate with the observation of a lower number of lung metastasis of CAIX-silenced 4T1 breast cancer cells in a murine model and also with that of reduced lung metastases of 4T1 cells expressing human CAIX with deleted PG-domain [16].

The importance of an acidic microenvironment for the proteolytic activity and extravasation was also documented *in vivo*. Intraperitoneally or orally introduced sodium bicarbonate reduces cathepsin- and MMP-based protease activity [33]. Bicarbonate administration neutralizes tumor acidity and even more, inhibits formation of spontaneous, as well as, experimental metastases [34]. Downregulation of MMP14 in MDA-MB-231 cells also reduced metastatic dissemination to the lungs [35].

Taken together, our data clearly show that CAIX regulates invadopodia formation by its enzymatic function, which influences proteolytic activity of matrix metalloproteases. The other consequence of CAIX depletion is decreased phosphorylation of cortactin (pY421) and reduced expression of Arp2, leading to attenuated actin free-barbed end formation and subsequent impairment of invadopodia growth. Finally, specific antibodies against proteoglycan or catalytic domain of CAIX reduce metastases in CAM and lung colonization murine model. These outcomes indicate the value of CAIX as a potential therapeutic target to overcome tumor progression.

## 4. Materials and Methods

### 4.1. Cell Culture

HT1080 and HeLa cells, and their transfected variants were maintained in DMEM (Lonza Biowhittaker (Basil, Switzerland), supplemented with 10% FCS (GE Healthcare Life Sciences) and Gentamicin (0.051 mg/mL) (Sandoz), and incubated at 37 °C with 5% CO_2_. Hypoxic experiments were conducted in the hypoxic workstation (Ruskinn Technologies) in 2% O_2_, 2% H_2_, 5% CO_2_, and 91% N_2_ atmosphere at 37 °C. Human esophageal squamous cell carcinoma (TE-1) cells were provided by Dr. Ari Ristimäki (Biomedicum, Helsinki, Finland). TE-1 cells were cultured in DMEM supplemented with 10% fetal bovine serum, 100 µg/mL streptomycin, and 100 U/mL penicillin at 37 °C with 5% CO_2_. Stable transfectants of C33-CAIX and C33-neo cells were prepared and maintained as described previously [36].

### 4.2. Transient Silencing

HT1080 and HeLa cells were seeded at 1.6 × 10^6^ cells in a 10 cm Petri dish. After 4h, cells were transfected with siCA9 and with control non-targeting siCtrl at 20 nM final siRNA concentration (siRNA smart pool system, Dharmacon (Lafayette, CO, USA). Transfection was performed according to the manufacturer’s recommendations (DharmaFECT siRNA Transfection Protocol, Thermo Scientific). The following day after transfection, cells were incubated in hypoxia (2% O_2_) for 24 h to trigger CAIX expression. Subsequently, cells were trypsinized, and seeded in a new Petri dish and further grown in the hypoxic chamber (2% O_2_) according to the experiment.

### 4.3. Immunoblotting

Post-incubation cells were washed with PBS and proteins were extracted for 15 min at 4 °C using RIPA lysis buffer (1% Triton X-100, 1% sodium deoxycholate, in PBS) supplemented with protease/phosphatase inhibitors. Samples containing 80 μg of total proteins were separated by 10% SDS-PAGE and immunoblotting was performed as described elsewhere [37]. Antibodies and dilutions were used as indicated in Table 1. All secondary antibodies (conjugated with horseradish peroxidase) were purchased from Dako. All western blot analysis was repeated 3 times.

### 4.4. Stable Transfection

HT1080 iRFP670 cells were prepared by a transfection with iRFP670-N1 plasmid (Addgene) using 6 μg of plasmid DNA and Xfect transfection reagent (Clontech) according to the manufacturer’s recommendations. Positive clones were selected with G418 (600 1000 μg/mL) for 2 weeks. Finally, the mixture of 8 positive clones was prepared.

### 4.5. Immunofluorescence

Cells grown on glass coverslips coated with collagen were fixed by 4% PFA in PBS for 10 min and permeabilized with 0.1% Triton X-100 in PBS for 5 min. The further procedure was performed as described in [14]. Samples were analyzed by Zeiss LSM510 Meta confocal microscope in the multitrack scanning mode. Primary antibodies were used as indicated in Table 1. Appropriate secondary antibodies (Invitrogen, Waltham, MA, USA) were used as follows: donkey anti-mouse Alexa Fluor 488, donkey anti-rabbit Alexa Fluor 488, Alexa Fluor 555 phalloidin for F-actin visualization.

### 4.6. Cell Fractioning

HT1080 and HeLa cells were cultured under hypoxic conditions (2% O_2_) for 24 h to induce CAIX expression. Subsequently, the cells were seeded at a sparse density on coverslips coated with gelatin (2 mg/mL) and left to invade the substrate for 24 h or 48 h in hypoxia (2% O_2_). Osmotic lysis was used for the separation of cell bodies (cytosol and membrane fraction) and invadopodial fraction as previously described [3].

### 4.7. Quantification of Invadopodia Formation via Cortactin Staining

HT1080 cells were transiently silenced by siCA9 or siCtrl, and pre-incubated in hypoxia for 24 h as described above. Then, the cells were seeded at 1 × 10^5^ on FITC-conjugated collagen (1 mg/mL) coated coverslips and left to attach, followed by 5h culture in hypoxia (2% O_2_). Samples were fixed with 4% PFA in PBS for 10 min, permeabilized with 0.1% Triton X-100 in PBS for 5 min and fluorescently labeled for cortactin as detailed above. Z-stack images were acquired by Zeiss LSM 510 Meta confocal microscope. The FITC-labeled collagen signal was used to determine the substrate layers which were processed further. Images were first thresholded by intensity to exclude background pixels. Only pixels with the positive cortactin signal were processed in ImageJ software (http://rsb.info.nih.gov/ij/). The number of cortactin spots and cortactin intensity were analyzed by the Analyze Particle tool; 23 cells were processed for each sample in total in two independent experiment.

### 4.8. Cell Invasion Assay

HT1080 cells were transiently silenced by siCA9 or siCtrl, pre-incubated in hypoxia for 24 h (2% O_2_) and starved in DMEM with 1% FCS overnight. Cell invasion was analyzed by xCELLigence RTCA DP instrument (Accela) in 16-well CIM-plate 16 (Accela) according to the manufacturer’s instructions. The cells were seeded, in quadruplicate, in a collagen-coated upper chamber (40,000 cells/well) in serum-free DMEM medium with hepatocyte growth factor (HGF, 20 ng/mL, Sigma). Cells invaded through the collagen and the microporous membrane into a lower chamber, which provided a chemotactic signal in the form of 10% FCS in DMEM. As cells adhere to the microelectrode sensor placed under the porous membrane, the electrical impedance increases and is measured in real-time by RTCA DP device. Relative changes in the electrical impedance reflect the invasion capacity of cells expressed as a dimensionless variable cell index.

### 4.9. Matrigel Degradation Assay

HeLa cells were transiently silenced by siCA9 or siCtrl, and pre-incubated in hypoxia for 24 h as described above. Afterwards, the cells were seeded at density 100,000 cells per 3.5 cm Petri dish with a glass bottom coated by Matrigel mixed with DQ-Red BSA (25 μg/mL, Invitrogen), providing a red fluorescent signal after its proteolytic cleavage within the ECM. After cell attachment, DMSO or the selective inhibitor of transmembrane isoforms of carbonic anhydrases HSFA (4-Homosolfanilamdie, Sigma A2134) was added, and cells were cultured for additional 24 h in hypoxia (2% O_2_). The final concentration of HSFA was 100 µM. The number, depth, and intensity of the red signal plaques reflecting invadopodia-mediated Matrigel digestion were analyzed by a Zeiss LSM 510 Meta confocal microscope. Acquired z-stack images were evaluated using the ImageJ program, Analyze Particles tool. Selected image slices positioned within Matrigel were intensity thresholded to exclude background pixels. Spots of digested matrix emitting the positive DQ-Red BSA signal were evaluated for their area and mean intensity values. In two independent experiments, 21 cells in total were analyzed for each sample.

### 4.10. Quail Chorioallantoic Membrane (CAM) Model

Fertilized Japanese quail (Coturnix japonica) eggs from a breeding colony (Laying Line 01, Institute of Animal Biochemistry and Genetics, Centre of Biosciences SAS) were incubated in a forced draught incubator at 37°C and 60% relative humidity. To prepare *ex ovo* culture, on the embryonic development day 3 (ED3) the surface of eggs was wiped with 70% ethanol in a sterile laminar flow hood. The eggs were opened and embryos transferred into six-well tissue culture plates (TPP, Switzerland) and returned to humidified incubator for the next 4 days [38]. Hypoxic tumor cells were implanted on ED7 CAM surface. After 72 h, CAMs with tumor cells were separated and fixed with 4% formaldehyde. Subsequently, 5 µm paraffin sections were prepared for imunohistochemical analysis.

### 4.11. Tumor Cells Implantation

For CAM assay, TE-1 cells were cultured in hypoxia (2% O2) for 48 h to induce CAIX expression. Cells were trypsinized, counted (2 × 10^6^ cells per sample), and in indicated samples, pre-incubated with a specific anti-CAIX antibodies Ab V/10 (Ab10) recognizing carbonic anhydrase domain of CAIX [39], or M75 recognizing proteoglycan domain of CAIX [9], (20 μg/mL) at the room temperature for 30 min. After treatment, cells were centrifuged at 0.9 rpm for 8 min and washed with PBS. Control samples were subjected to the same procedure. Cells implantation was performed on ED7 under sterile conditions. A silicone ring (6 mm) was positioned on the CAM surface and the cell suspension was placed therein.

### 4.12. Immunohistochemistry

Formalin or formaldehyde fixed, paraffin-embedded tumor tissues were cut into 5 µm sections. The immunohistochemical staining was performed as described by Takacova and colleagues [40] using DakoCytomation EnVisio+ System-HRP (DAB). CAIX was stained by M75 antibody (hybridoma medium diluted 1:100) for 1 h at room temperature without antigen retrieval. Pictures were taken by a Leica DM4500B microscope with a Leica DFC480 camera.

### 4.13. In Vivo Model of Experimental Metastasis

NMRI nude mice (NMRI-Foxn1nu nu/nu female mice, Charles River Laboratories, Inc.) were housed and used in accordance with the Institutional Ethic Committee guidelines under the approved protocols. The project was approved by the national competence authority – State Veterinary and Food Administration of the Slovak Republic (No. Ro. 4245/13-221 and 292/16-221g, validity 1/1/2014−31/12/2017 and 1/1/2014−31/12/2018, respectively) in compliance with the Directive 2010/63/EU and the Regulation 377/2012 on the protection of animals used for scientific purposes. Hypoxia pre-incubated HT1080 cells (1% O_2_, 40 h) expressing red fluorescent protein (HT1080-RFP) were used for tail vein colonization assay. Cells were trypsinized and in indicated samples were pre-incubated with Ab10 or M75 antibody (20 μg/mL) for 30 min at the room temperature. After centrifugation at 0.9 rpm for 5 min at 4^◦^C and washing with PBS, the cells were counted and diluted to a total of 200 μL PBS containing 1.5 × 10^6^ cells. Control cells were subjected to the same handling. Cells in PBS were injected into the tail vein of nude mice divided into groups (Control, Ab10 and M75) of 10 animals each. During the experiment, the antibodies were intravenously administered (50 μg in 100 μL PBS) twice to ensure a permanent coverage of the desired epitope. After 12 days mice were sacrificed. Mice were anesthetized and their lungs were perfused with PBS and 10% formalin. Excised lungs were evaluated for fluorescent signal emitted by metastasizing cancer cells using an IVIS system (*In vivo* Imaging System, Caliper Life Sciences). Image sequences were acquired for every pair of lungs (bottom and upper side) at 14 combinations of excitation and emission filters, at field of view B (FOV B). Spectral unmixing was performed to separate the specific fluorescent signal emitted by the fluorophore and tissue autofluorescence. All lungs were marked as separate ROIs (Region of Interest), and the fluorescent signal was integrated over the whole ROI area and expressed as total radiant efficiency. This variable is normalized and allows the comparison between different images. The average of both values (bottom and upper view) of the total radiant efficiency was used as the final result for each lung.

## 5. Conclusions

In summary, we have confirmed our previous findings that CAIX enhances cancer cell invasion via a pH-dependent mechanism and its proteoglycan domain. Furthermore, we have demonstrated for the first time the localization of CAIX in invadopodia *in vivo*. We also show that CAIX modulates molecular signaling coordinating actin polymerization within invadopodia through its effect on cortactin phosphorylation at Y421 as well as on the expression of actin-associated proteins. Our results give evidence of the essential role of CAIX in extracellular matrix cleavage during cell invasion and highlight its potential to be the target in antitumor therapy.

## Figures and Tables

**Figure 1 ijms-20-02745-f001:**
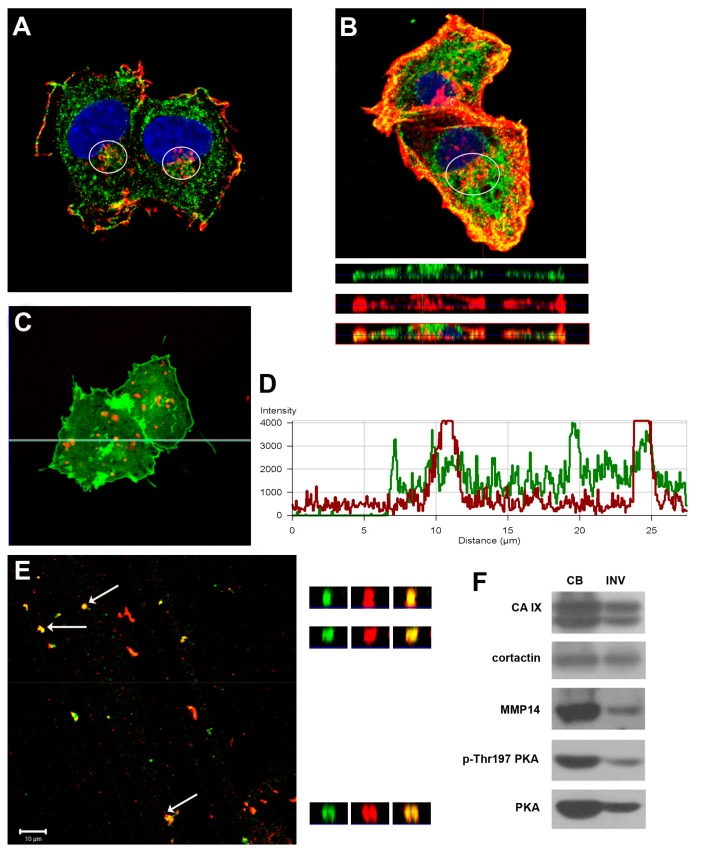
CAIX is present in active invadopodia and colocalizes with invadopodial marker cortactin. (**A**) Immunofluorescent analysis of hypoxia pre-incubated HT1080 cells shows colocalization of CAIX (green) and cortactin (red) in the perinuclear area, marked by a white ellipse, within only 5 h after cells seeding onto a collagen layer. (**B**) CAIX also colocalizes with F-actin (red) in protruding invadopodium as documented by z-stack analysis of cells cultured for 24 h on collagen. (**C**) C33 cells transfected with CAIX-GFP protein were cultured on the Matrigel layer containing DQ-BSA as a marker of proteolytic activity (red). Panel (**D**) shows the intensity profile of CAIX and DQ-BSA signals along the green line denoted in (**C**). Overlapping peaks of green CAIX signal with red peaks indicating spots with proteolytic degradation of the matrix. (**E**) After osmotic lysis of HT1080 cells cultured on gelatin for 48 h in hypoxia the invadopodia entrapped within a gelatin layer were visualized by immunofluorescence (CAIX – green, cortactin - red). The right panels show xz-sections of invadopodia, indicated by white arrows, across the matrix depth. (**F**) Representative Western blot analysis of cell body fraction and isolated invadopodia of HT1080 cells confirms the presence of CAIX in the invadopodial fraction, together with the active form of PKA which phosphorylates CAIX.

**Figure 2 ijms-20-02745-f002:**
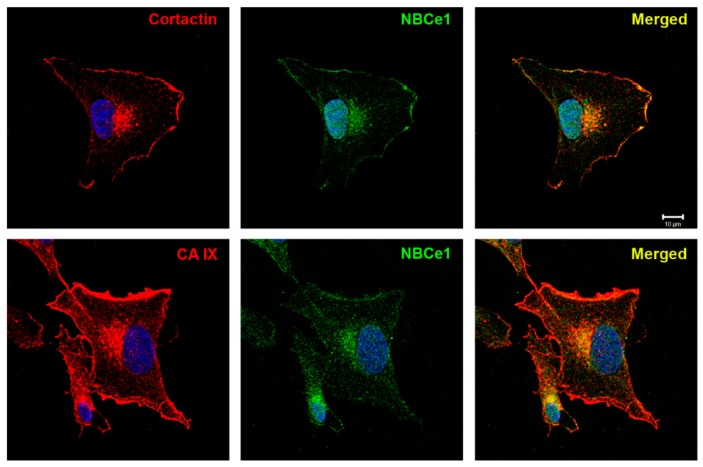
CAIX colocalizes with bicarbonate transporter NBCe1 in HT1080 cells seeded on collagen. After 5h of invadopodia assembly in 2% hypoxia, the samples were stained for NBCe1 and cortactin (**A**) and NBCe1 together with CAIX (**B**). Nuclear DAPI staining is blue. Confocal microscopy analysis revealed the occurrence of bicarbonate transporters within invading cells and their colocalization with CAIX and the invadopodial marker cortactin. The coincident distribution of both mentioned protein pairs was observed in the close proximity to the nucleus.

**Figure 3 ijms-20-02745-f003:**
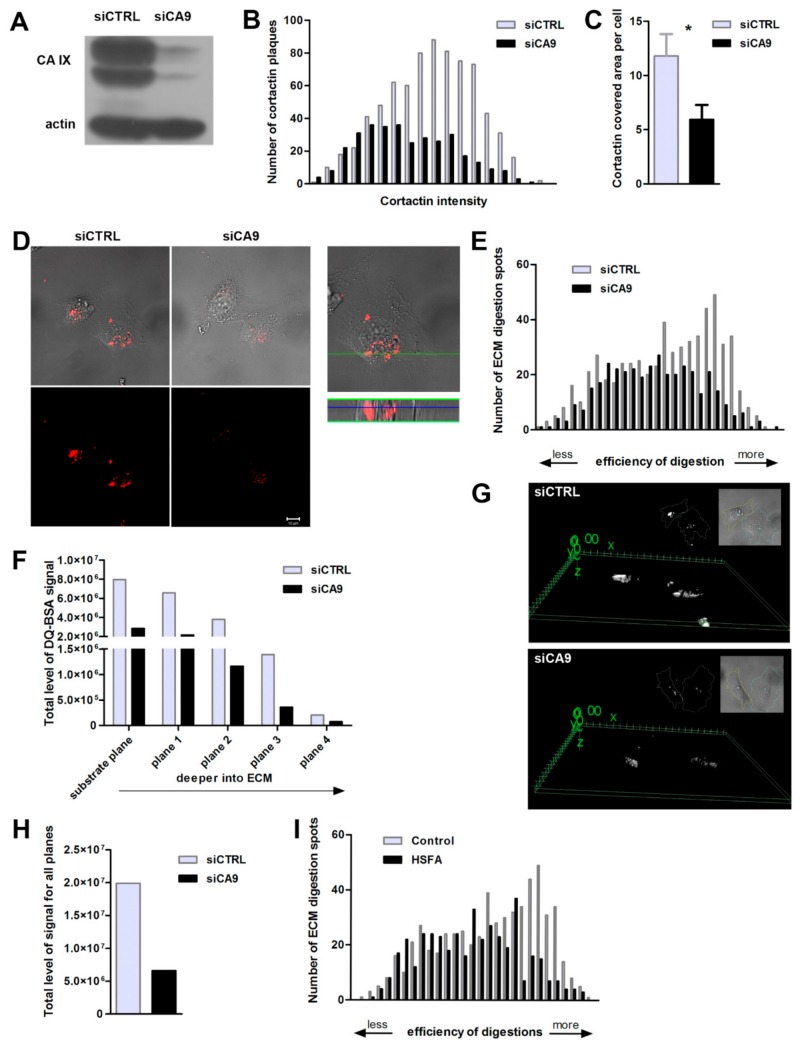
CAIX expression promotes invadopodia formation and increases the efficiency of extracellular matrix digestion. (**A**) Representative Western blot proving a high efficiency of transient CA9 silencing in hypoxia-cultured HT1080 cancer cells. (**B**) Hypoxic CA9-attenuated (siCA9) and control CAIX expressing (siCTRL) HT1080 cells were grown for 5 h on FITC-collagen which allowed us to define the plane of the substrate. Samples were fixed and fluorescently stained for cortactin (N = 23 cells, in two independent experiments). CAIX expressing cells (siCTRL) show a higher number of cortactin-containing invadopodia characterized by stronger fluorescent intensity in comparison to CA9-suppressed cells (siCA9). These results indicate a stimulating role of CAIX in the cortactin accumulation during the invadopodia precursor assembly. (**C**) Analysis of the overall area covered by cortactin-containing invadopodia in individual cells. The graph shows mean ± stdev of cortactin-stained area fractions, calculated as a percentage of the total cell area. Cell areas were determined according to corresponding transmitted light (DIC) images. The area covered by cortactin-stained invadopodial plaques was significantly reduced in CA9-silenced cells (*t*-test, * *p* ≤ 0.05). (D-I) Hypoxic HeLa cells were seeded on the Matrigel layer mixed with DQ-BSA providing a red signal after active invadopodia-mediated focalized digestion of ECM. (**D**) Representative images of control (siCTRL) and CA9-attenuated cells (siCA9) taken with (upper row) and without transmitted light (bottom row) indicating impaired Matrigel cleavage by CA9-suppressed cells. Right-hand panel shows zoomed-in area of ECM digestion in a control cell. A green line indicates the position of z-section across the matrix depth displayed below. (**E**) Histogram of numbers of matrix degradation spots sorted by the intensity of DQ-BSA signal, reflecting efficiency of invadopodial proteolytic activity, in CAIX expressing and CA9 silenced cells (*N* = 21 for each sample in two independent experiments). (**F**) Graph shows the total level of positive fluorescent signal originating from digested matrix for all 21 CAIX expressing and CA9-depleted cells. Data were processed separately for each plane, from the substrate plane (0) to planes located deeper inside the Matrigel (1 to 4). The signal evaluation includes the overall area of digestion as well as the intensity of digested spots. (**G**) 3D model of digested matrix signal (DQ-BSA) created for representative images of CAIX expressing (upper panel) and CA9-attenuated cells (lower). The modeling was done in 3D Viewer plugin in ImageJ from substacks of ECM planes, where digestion takes place. Cell outlines were drawn according to corresponding DIC images of cells. (**H**) Total level of digested matrix signal summed up for all planes for all processed cells confirming a higher efficiency of Matrigel digestion of CAIX positive cells. (I) Histogram of numbers of matrix digestion spots sorted by the intensity of DQ-BSA signal comparing cells (*N* = 21) treated with HSFA, which inhibits catalytic activity of CAIX, and cells with fully active CAIX (DMSO, *N* = 21).

**Figure 4 ijms-20-02745-f004:**
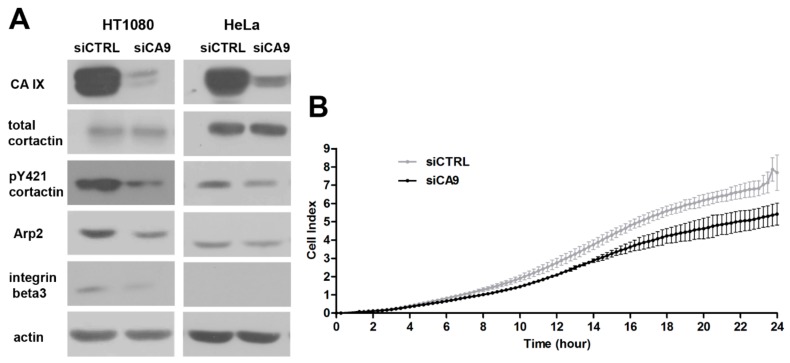
CA9 silencing affects the protein level of invadopodia-associated molecules and influences cancer cell invasion (**A**) Representative Western blot analysis of hypoxia pre-incubated HT1080 and HeLa cells transiently silenced by siCA9. Cells were seeded on gelatin and cultured in 2% hypoxia for further 24 h. CA9 silencing led to reduced protein levels of Arp2 and a phosphorylated form of cortactin in both cell lines. Total level of cortactin remained unchanged. In HT1080, CA9 downregulation also resulted in a decreased level of integrin beta3. This type of integrin was not detected in HeLa. Arp2 and integrin beta3 are involved in invadopodia formation; phosphorylation of Y421 at cortactin is essential for maturation of invadopodia precursors. (**B**) Invasion ability of hypoxia pre-incubated HT1080 cells expressing CAIX and with CA9 silenced was assessed using real-time measurement by xCELLigence device. Cells were seeded in a collagen-coated Boyden chamber, under HGF treatment, and stimulated into invasion toward a chemoattractant in the lower chamber. CAIX expressing cells invade faster than their counterparts without CAIX as shown by higher cell index values, which reflect the number of cells that penetrated through a collagen layer. The graph shows time dependence of the cell index expressed as mean±stdev of quadruplicates. The experiment was repeated three times.

**Figure 5 ijms-20-02745-f005:**
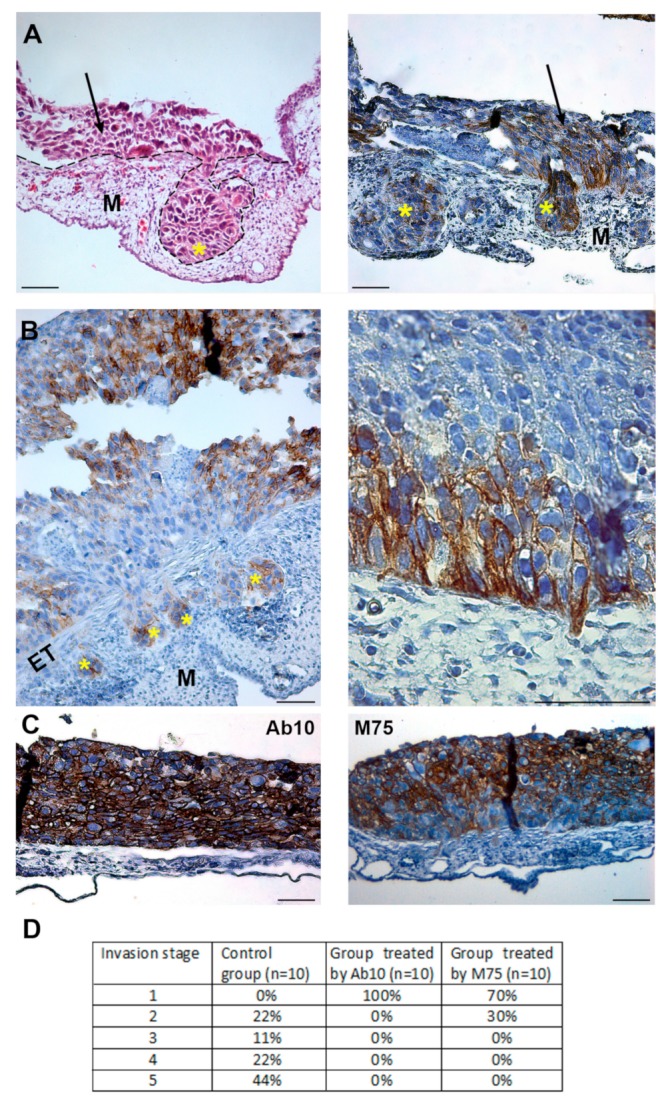
The effect of CAIX on invasion of hypoxic tumor cells in the quail chorioallantoic membrane (CAM) model. Hypoxia pre-incubated TE-1 cells (48 h, 1% O_2_) were grafted onto the day 7 quail embryo and cultivated *ex ovo* for 3 days. Hematoxylin and Eosin staining, and imunohistochemical staining of CAIX. (**A**) Hypoxic tumor cells (outlined by dashed line) invaded into CAM and formed metastatic foci (marked by asterisks) in the mesoderm (M) which are positive for CAIX staining. Left image: hematoxylin and eosin staining, right image: immunostaining of CAIX. Tumor cells adhering on the ectoderm layer of CAM are indicated by arrows. (**B**) Immunostaining of CAIX of control, untreated cells invading into the mesoderm (M). Left image: clusters of migrating cells display CAIX localization at the invasive front (denoted by asterisks). Right image: distribution of CAIX into invadopodium penetrating the ectoderm layer. (**C**) Effect of anti-CAIX antibodies on invasion ability of TE-1 cells. Hypoxia pre-incubated TE-1 cells were treated by anti-CAIX antibodies before the seeding on CAM. Ab10 antibody targeting the carbonic anhydrase domain completely inhibited the metastatic properties of cancer cells. Anti-PG antibody (M75) strongly suppressed the invasive capability of TE1 cells. Scale bars 50 µm. Ectoderm layer of CAM (ET), mesoderm (M). (**D**) The effect of CAIX targeting antibodies on hypoxic tumor cell invasion into CAM. The invasion process was divided into five stages. Increasing numbers denote gradually advancing invasion. The stages 1 to 5 are defined as follows: (1) no cells invading into CAM; (2) only a few invading cells which penetrate the ectoderm layer; (3) cells started invading the mesoderm; small clusters are being formed; (4) cells invaded into the mesoderm, metastatic foci had already been formed, (5) considerable metastasis throughout the whole CAM structure. Five CAMs with grafted cells were assessed for each sample during two independent experiments. The values in the table give the portions (in %) of all evaluated CAMs (from both experiments) allocated to separate stages.

**Figure 6 ijms-20-02745-f006:**
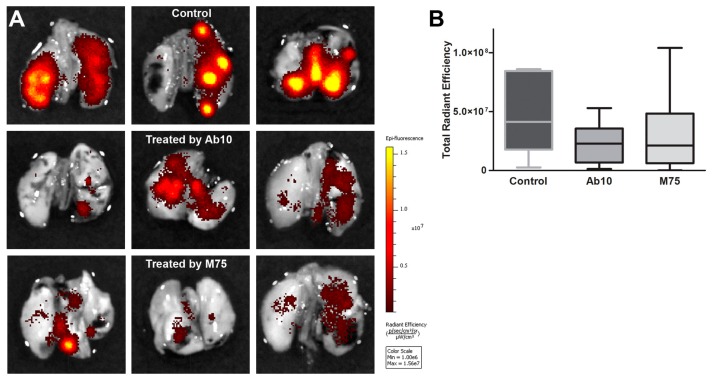
Treatment with CAIX-targeting antibodies prevents extensive lung metastasis formation during *in vivo* colonization assay. (**A**) Representative ex vivo images of fluorescent lung metastases of control mice (top panel), of mice treated with Ab10 antibody against the catalytic domain of CAIX and mice treated with M75 antibody against CAIX proteoglycan domain. Ex vivo imaging was performed 12 days after inoculation of RFP-labeled HT1080 cells using IVIS Caliper. Different stages of lung infiltration by HT1080-RFP cells correspond to the intensity and extent of the fluorescent signal. Scale of fluorescence intensity is shown on the right side of the picture with yellow color representing the strongest emission. (**B**) Box plot graph shows signal distribution of all analyzed lungs based on total radiant efficiency values. Higher total radiant efficiency values mean higher number of fluorescence emitting cells metastasizing into lungs. The lines denote median values. Control mice show a trend toward a higher lung signal than antibody-treated mice. These results indicate that cells expressing fully active CAIX protein with no domain blockage are more likely to succeed in extravasation and metastasis initiation than their counterparts treated with antibodies against CAIX.

**Figure 7 ijms-20-02745-f007:**
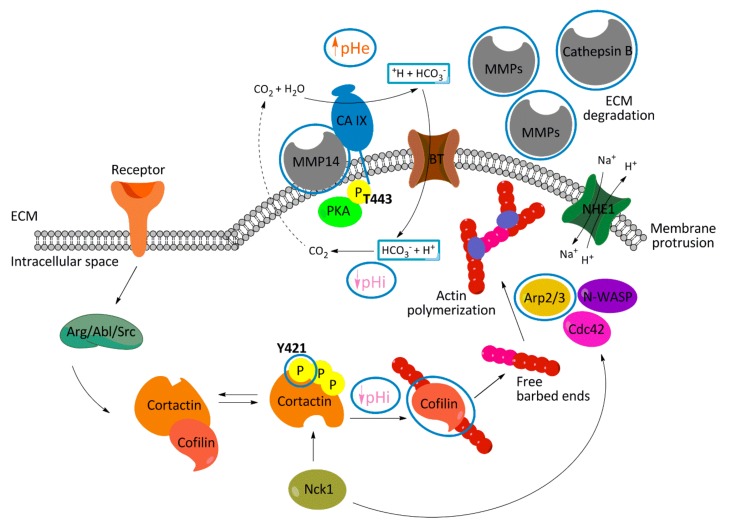
Hypothetical model depicting the consequences of downregulation of CAIX expression (denoted by blue circles) on mechanisms that modulate processes required for the invadopodia development. Acidic extracellular pH is indispensable for invadopodia formation, proteolytic cleavage of extracellular matrix and cancer cell invasion. Enzymatic activity of CAIX regulated by protein kinase A (PKA) generates acidic pHe nanodomains promoting the activity of MMPs. The functional cooperation between CAIX and bicarbonate transporters (NBCe1) in invadopodia locally increases pHi and releases pH-dependent cortactin-cofilin binding to promote actin-free barbed end formation. When CAIX is depleted, extracellular pH increases, intracellular pH becomes less alkaline, phosphorylation of Y421-cortactin is reduced, and Arp2 expression is lowered. These CAIX-related changes attenuate invadopodia formation and actin polymerization. Altogether, CAIX-mediated pH regulation as well as CAIX expression enhances proteolytic activity in invadopodia and affects actin-regulating proteins essential for invadopodia elongation.

**Table 1 ijms-20-02745-t001:** List of primary antibodies.

Antigen	Host	Methodology–Dilution	Company
actin	g	WB - 1:1000	Santa Cruz
anion exchanger 2 (AE2)	r	IF - 1:500	GeneScript–on request [14]
Arp2	r	WB - 1:1000	Santa Cruz (sc-15389)
carbonic anhydrase IX (CAIX)	m	WB - 1:3 (hybridoma medium) IF - 1:250 (M75-AlexaFluor 488 conjugate) IHC - 1:100	in house, M75 antibody [9]
cortactin	m	WB - 1:1000 IF - 1:100	Millipore (05-180)
F-actin (phalloidin AF-555)	-	IF - 1:40	Invitrogen (A34055)
integrin beta 3	r	WB - 1:1000	Abcam (ab75872)
Ki-67	m	IHC - 1:100	DAKO (M7240)
matrix metalloproteinase 14 (MMP14)	r	WB - 1:1000	Millipore (AB6004)
electrogenic sodium bicarbonate cotransporter 1 (NBCe1)	r	IF - 1:100	Millipore (AB3212)
paxillin	r	IF - 1:250	Santa Cruz (sc-5574)
phospho-cortactin	r	WB - 1:1000	Cell Signaling (4569)
phospho- protein-kinase A (p-PKA)	r	WB - 1:1000	Cell Signaling (5661S)
protein-kinase A (PKA)	r	WB - 1:1000	Cell Signaling (4782S)

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
