# Peer review of "CAIX Regulates Invadopodia Formation through Both a pH-Dependent Mechanism and Interplay with Actin Regulatory Proteins"

_ijms, 2019, doi:10.3390/ijms20112745_

Round 1

Reviewer 1 Report

Abstract:  This is a manuscript describing a functional relationship between CA-IX, NBC, and the contactin-cofilin complex to promote invadopodially mediated invasion.  The primary analytic methods were quantitative ICC.  Knockdown or inhibition of CA-IX inhibited invadopodia formation, contacting phosphorylation, and matrix degradation.  Inhibition of CA-IX with directed antibodies Ab10 or M75 inhibited metastases in CAM and lung colonization models.

Comments

1.     Need ref for HSFA.  Also, what was final concentration?  Assume it is not 0.5 mM.

2.    Comment on specificity of the Antibody used for NBCe1 versus NBCn1.  

3.    Figure 7. pHe decreases and pHi increases, not the other way around as shown.  

4.    Antibody table in methods should provide catalog number.

Reviewer 2 Report

Debreova et al “CAIX Regulates Invadopodia Formation through both pH-Dependent Mechanism and Interplay with Actin Regulatory Protein”

The work by Debreova and colleagues investigates the mechanism by which CA9 could stimulate cell invasion contributing to metastasis.  They report localization of a portion of cellular CA9 at invadopodia and provide evidence that this a functional localization that stimulates the activity of these organelles.  The findings are interesting, and the work is generally well done.  The observations are significant, but some additional clarifications would help to support their conclusions.

Major Points.

1.      Very little statistical analysis is provided in the figures or text.  In figure 1, how many cells/slices were analyzed?  How often did regions of GFP positivity colocalize with red fluorescence shown in figure 1D?  There appears to be 4 regions of GFP, bot only 2 show significant proteolysis.  In figure 3B C E F I, how many cells/plaques were counted? In figure 5D how many total cells/fields were counted?

2.      In figure 5, it is difficult to distinguish the tumor cells from the quail cells.  Could you try using labelled tumor cells to make this more obvious?  This is a nice assay and result, so making it easy to see would be helpful (ie like the RFP labelled HT1080 cells form figure 6).

3.      In figure 6, no statistical significance is claimed in the amount of tumor growth in the lungs of the treated versus untreated mice. Is there a difference?  How does this compare to lung colonization of cells with CA9 stably knocked down? This appears to be important for the paper’s conclusion. 

4.      Why were the different cell lines chosen for the different experiments? No rational is presented in the manuscript.  Some consistency would be helpful to convince the reader (ie use the line with the best correlation between CA9 expression and other markers of invadopodia like HT1080 cells shown in figure 4).  However, it is good to see more than one cell line used overall.

5.      Why was NBCe1 chosen?  There are many bicarbonate transporters, several of which are regulated by hypoxia.  Could others be contributing as well?

Minor points.

1.      Why was the axis broken in figure 3F?  It seems unnecessary.

The discussion seems a bit long for the manuscript

Author Response

Response to Reviewer 2 Comments

Reviewer 2

The work by Debreova and colleagues investigates the mechanism by which CA9 could stimulate cell invasion contributing to metastasis.  They report localization of a portion of cellular CA9 at invadopodia and provide evidence that this a functional localization that stimulates the activity of these organelles.  The findings are interesting, and the work is generally well done.  The observations are significant, but some additional clarifications would help to support their conclusions.

We would like to thank to the reviewer for his/her valuable comments. Please, see our responses below. The changes made in the text are marked as revisions or clearly described in the answers given below.

Major Points.

Point 1: Very little statistical analysis is provided in the figures or text.  In figure 1, how many cells/slices were analyzed?  How often did regions of GFP positivity colocalize with red fluorescence shown in figure 1D?  There appears to be 4 regions of GFP, bot only 2 show significant proteolysis.  In figure 3B C E F I, how many cells/plaques were counted? In figure 5D how many total cells/fields were counted?

Response 1: In Fig 1 we wanted to present data showing that CAIX is able to localize into invadopodia. Therefore, we used several combinations of invadopodia markers with CAIX to demonstrate their colocalization or colocalization of CAIX spots with areas matrix degradation indicated by DQ-BSA. Fig. 1 A, B, C, E give representative images of these experiments, in which tens of cells were examined. The important proof of CAIX localization within invadopodia is given the analysis of isolated invadopodia trapped within matrix after osmotic lysis. The isolated invadopodia content was assessed also by western blot, and CAIX was detected within invadopodia together with invadopodia markers cortactin, and MMP14. Fig 1D shows the course of intensities of green (CAIX) and red (DQ-BSA) signals along the line denoted in the figure 1C. We marked this line more clearly in white color???

To quantitatively assess the impact of CAIX expression and function on proteolytic degradation of ECM we used CA9 silencing and enzymatic inhibition. Data are presented in Fig.3. To evaluate the impact of CA9 silencing on invadopodia formation 23 cells were analyzed for each sample (i.e. silenced and control cells), over 1000 cortactin spots were analyzed altogether. To evaluate the impact of CA9 silencing on efficiency of matrix digestion 21 cells were analyzed for each sample, five planes (z-stack slices) were assessed for each cell, nearly 3000 spots were analyzed altogether. To assess the effect of CAIX inhibition by HSFA on matrix digestion 21 cells were analyzed for each sample (control vs HSFA treated), five planes for each cell, approx. 3000 spots altogether.

As for Fig 5D: We apologize for unclear description of the table. It summarizes the extent of the invasion of tumor cells, grafted onto CAM, into the subsequent CAM layers (ectoderm, mesoderm). Cells were either treated with antibodies against CAIX or left untreated as the control. The invasion process was divided into 5 stages described in the figure legend. We changed the table headings to make the results clearer. The percentage values in the table show in how many cases (out of 10) the cells seeded on CAM managed to penetrate within it. Ab10 completely abrogated this process in all tested cases (cells seeded on 10 CAMs did not manage to invade any of them).

Point 2: In figure 5, it is difficult to distinguish the tumor cells from the quail cells.  Could you try using labelled tumor cells to make this more obvious?  This is a nice assay and result, so making it easy to see would be helpful (i.e. like the RFP labelled HT1080 cells form figure 6).

Response 2: We added explanatory markings into the Fig. 5 to better distinguish grafted tumor cells and separate CAM layers. Labeling of tumor cells by RFP would present some difficulties as we would have to use fluorescence detection. Alternatively, we could try to employ antibody against RFP in immunohistochemical analysis which is standardly used to assess CAM assays; we will consider such approach in our future experimental work.

Point 3: In figure 6, no statistical significance is claimed in the amount of tumor growth in the lungs of the treated versus untreated mice. Is there a difference?  How does this compare to lung colonization of cells with CA9 stably knocked down? This appears to be important for the paper’s conclusion. 

Response 3: Outcome of the lung colonization assay was analyzed using ex vivo measurement of total radiant efficiency by IVIS which enables the evaluation of the signal from the whole lung. Our results show differences in geometric mean: 3.49x107, 1.5x107, and 2.31x107 for Control, Ab10 treated and M75 treated groups respectively (i.e. reduction to 42% (Ab10) and 66% (M75) of the control) and also differences in median values: 4.13x107, 2.28x107, and 2.68x107 for Control, Ab10 treated and M75 treated groups (i.e. signal decrease to 55% and 65% of the Control). Differences are not statistically significant due to rather high scatter of measured fluorescent signal values which is common in animal experiments. We changed the graph type in Fig. 6B to a box plot (instead of scatter plot with logarithmic axis) to better present the data and differences among the sets. We also performed IHC analysis of lungs sections stained with the proliferation marker Ki-67 to visualize tumor cells which successfully penetrated inside lungs and formed metastatic foci. Please, see sets of representative images of lungs sections below. The amount of IHC-stained metastasizing cells corresponds to the level of fluorescent signal emitted by HT1080-RFP cells measured by IVIS in lungs. The images confirm that treatment by antibody results in reduction of lung colonization by tumor cells.

When the whole CAIX protein is knocked down the effect on lungs colonization is more pronounced than in case that only PG domain is deleted as published in Swayampakula et al (2017). We chose the approach using human tumor cells and specific anti-CAIX antibodies prepared in our laboratory. The body of evidence exist showing that CAIX is also expressed in circulating tumor cells. This fact makes it a relevant therapeutic target during anti-metastatic treatment. Our results indicate that antibodies against CAIX could help reduce metastasis formation.

Point 4: Why were the different cell lines chosen for the different experiments? No rational is presented in the manuscript.  Some consistency would be helpful to convince the reader (ie use the line with the best correlation between CA9 expression and other markers of invadopodia like HT1080 cells shown in figure 4).  However, it is good to see more than one cell line used overall.

Response 4: Acidity of tumor microenvironment and the importance of pH regulation in migration and invasion of tumor cells is a general phenomenon in which CAIX plays the essential role. It is well known that CAIX is broadly expressed in a different carcinoma types e.g glioblastoma, breast, fibrosarcoma, lung, cervix uteri, colorectal etc. (Pastorekova & Zavada, Cancer Therapy, 2004). Thus, we used several tumor cell lines during our investigations to confirm that CAIX involvement in invadopodia is not limited to specific cell line and is valid across a range of cell lines of different origin.

Point 5: Why was NBCe1 chosen?  There are many bicarbonate transporters, several of which are regulated by hypoxia.  Could others be contributing as well?

Response 5: In our previous work, we proved the metabolon formation between CAIX and NBCe1 in the lamellipodia of migrating cells (Svastova et al., 2012). This cooperation maximizes efficiency of pH gradient generation. As pH regulation is also important for invadopodia formation and maturation we wanted to investigate if this metabolon is also formed within invadopodia. We expect that other bicarbonate transporters could also contribute this process.

Minor points.

Point 1: Why was the axis broken in figure 3F?  It seems unnecessary.

Response 1: We wanted readers to better see a difference in DQ-BSA signal between CA9 silenced and control cells at planes 3 and 4 which lay deeper within ECM.

The discussion seems a bit long for the manuscript
